# Women Wearing Lipstick: Measuring the Bias Between an Object and Its Related Gender

**Ahmed Sabir**    **Lluís Padró**

Universitat Politècnica de Catalunya, TALP Research Center, Barcelona, Spain

asabir@cs.upc.edu, padro@cs.upc.edu

## Abstract

In this paper, we investigate the impact of objects on gender bias in image captioning systems. Our results show that only gender-specific objects have a strong gender bias (*e.g.* *women-lipstick*). In addition, we propose a visual semantic-based gender score that measures the degree of bias and can be used as a plug-in for any image captioning system. Our experiments demonstrate the utility of the gender score, since we observe that our score can measure the bias relation between a caption and its related gender; therefore, our score can be used as an additional metric to the existing Object Gender Co-Occ approach. Code and data are publicly available at https://github.com/ahmedssabir/GenderScore.

## 1 Introduction

Visual understanding of image captioning is an important and rapidly evolving research topic (Karpathy and Fei-Fei, 2015; Anderson et al., 2018). Recent approaches predominantly rely on Transformer (Huang et al., 2019; Cho et al., 2022) and the BERT based pre-trained paradigm (Devlin et al., 2019) to learn cross-modal representation (Li et al., 2020; Zhang et al., 2021; Li et al., 2022, 2023).

While image captioning models achieved notable benchmark performance in utilizing the correlation between visual and co-occurring labels to generate an accurate image description, this often results in a gender bias that relates to a specific gender, such as confidently identifying a woman when there is a kitchen in the image. The work of (Zhao et al., 2017) tackles the problem of gender bias in visual semantic role labeling by balancing the distribution. For image captioning, Hendricks et al. (2018) consider ameliorating gender bias via a balance classifier. More recently, Hirota et al. (2022) measured racial and gender bias amplification in image captioning via a trainable classifier on an additional human-annotated existing caption dataset for the gender bias task (Zhao et al., 2021).

To close the full picture of gender bias in image captioning, in this work, unlike other works, we examine the problem from a visual semantic relation perspective. We therefore propose a Gender Score via human-inspired judgment named Belief Revision (Blok et al., 2003) which can be used to (1) discover bias and (2) predict gender bias without *training* or *unbalancing* the dataset. We conclude our contributions are as follows: (1) we investigate the gender object relation for image captioning at the word level (*i.e.* object-gender) and the sentence level with captions (*i.e.* gender-caption); (2) we propose a Gender Score that uses gender in relation to the visual information in the image to predict the gender bias. Our Gender Score can be used as a plug-in for any out-of-the-box image captioning system. Figure 1 shows an overview of the proposed gender bias measure for image captioning.

## 2 Visual Context Information

In this work, we investigate the relation between gender bias and the objects that are mainly used in image captioning systems, and more precisely, the widely used manually annotated image caption datasets: Flickr30K (Young et al., 2014) and COCO dataset (Lin et al., 2014). However, we focus mainly on the COCO dataset as the most used dataset in gender bias evaluation. COCO Captions is an unbalanced gender bias dataset with a 1:3 ratio bias towards men (Zhao et al., 2017; Hendricks et al., 2018; Tang et al., 2021). The dataset contains around 120K images, and each image is annotated with five different human-written captions.

To obtain the *visual context o* from each image $I$, we use out-of-the-box classifiers to extract the image context information $o(I)$. Specifically, following (Sabir et al., 2023), the objects extracted from all pre-trained models are obtained by extracting the top-3 object class/category (excluding *person* category) from each classifier after filtering out instances with (1) the cosine distance between

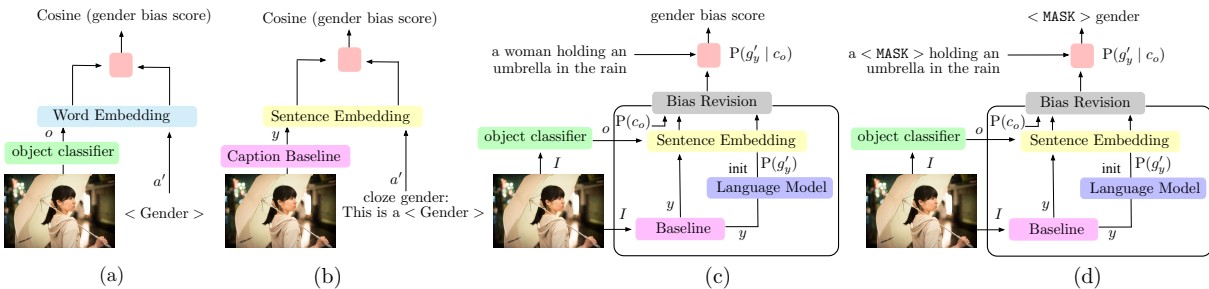

Figure 1: An overview of the proposed gender bias measure for image captioning. (a) word level with word embedding, (b) sentence level with semantic embedding, (c) our proposed gender score that relies on visual bias revision (with GPT-2 for initialization $\mathrm{P}\left(g'_y\right)$ (common observation as initial bias without visual) and with BERT for similarity score), and (d) the proposed < MASK > gender estimation (prediction), also with visual bias revision.

objects (voting between classifiers of top-$k$) and (2) a low confidence score via a probability threshold $< 0.2$. We next describe each of these classifiers:

| GloVe Gender Dist (Biased) | | | GloVe Gender Dist (Balanced) | | |
|---|---|---|---|---|---|
| + person | + man | + woman | + person | + man | + woman |
| face | walking | lipstick | can | police | police |
| hand | cowboy | dishwasher | police | white | black |
| ballplayer | gorilla | sleeping | go | walking | white |
| mailbox | teddy | horse | walking | black | car |
| book | ballplayer | bathtub | black | car | walking |

Table 1: Most frequency count of object + gender in the COCO Captions training dataset via Cosine Distance with GloVe and balanced Gender Neutral GN-GloVe.

**ResNet-152 (He et al., 2016):** A residual deep network that is designed for ImageNet classification tasks, which relies heavily on batch normalization.
**CLIP (Radford et al., 2021):** (Contrastive Language-Image Pre-training) is a pre-trained model with contrastive loss where the pair of image-text needs to be distinguished from randomly selected sample pairs. CLIP uses Internet resources without human annotation on 400M pairs.
**Inception-ResNet FRCNN (Huang et al., 2017):** is an improved variant of Faster R-CNN, with a trade-off of better accuracy and fast inference via high-level features extracted from Inception-ResNet. It is a pre-trained model that is trained on COCO categories, with 80 object categories.

## 3 Gender Object Distance

The main component of our Gender Score (next section) is the semantic relation between the object and the gender. Therefore, in this section, we investigate the semantic correlation between the object and the gender in the general dataset (*e.g.* wiki). More specifically, we assume that all images contain a gender *man*, *woman* or gender neutral *person*, and we aim to measure the Cosine Distance between the object $o$, objects with context

*i.e.* caption $y$ from the image $I$ and its related gender $a' \in \{$man, woman, *person*$\}$. In this context, we refer to the Cosine Distance as the closest distance (*i.e.* semantic relatedness score) between the gender-to-object via similarity measure *e.g.* cosine similarity. We apply our analysis to the most widely used human-annotated datasets in image captioning tasks: Flikr30K and COCO Captions datasets.

We employ several commonly used pre-trained models: (1) word-level: GloVe (Pennington et al., 2014), fasttext (Bojanowski et al., 2017) and word2vec (Mikolov et al., 2013) out-of-the-box as baselines, then we utilize Gender Neutral GloVe (Zhao et al., 2018), which balances the gender bias; (2) sentence-level: Sentence-BERT (Reimers and Gurevych, 2019) tuned on Natural Language Inference (NLI) (Conneau et al., 2017), (2b) SimCSE (Gao et al., 2021) contrastive learning supervised by NLI, and an improved SimCSE version with additional **Info**rmation aggregation via masked language model objective InfoCSE (Wu et al., 2022). In particular, we measure the Cosine Distance from the gender $a'$ to the object $o$ or caption $y$ as shown in Figure 1 (a, b) (*e.g.* how close the gender vector of $a'$ is to the object $o$ *umbrella*). Table 1 shows the top most object-gender frequent count with the standard and gender-balanced GloVe, respectively.
**Gender Bias Ratio:** we follow the work of (Zhao et al., 2017) to calculate the gender bias ratio towards men as:

$$\text{bias}_{\text{to-}m} = \frac{\text{s(obj}, m)}{\text{s(obj}, m) + \text{s(obj}, w)} \quad (1)$$

where $m$ and $w$ refer to the gender in the image, and the $s$ is our proposed gender-to-object semantic relation bias score. In our case, we also use the score to compute the ratio to gender neutral *person*:

$$\text{Ratio}_{\text{to-}n} = \frac{\text{s(obj}, m/w)}{\text{s(obj}, person)} \quad (2)$$

|  | COCO Captions | | | | | Flikr30K | | | | |
|  | Avg: Gender Object Distance | | | Ratio | | Avg: Gender Object Distance | | | Ratio | |
| Model | + person | + man | + woman | to-m | to-w | + person | + man | + woman | to-m | to-w |
|---|---|---|---|---|---|---|---|---|---|---|
| Word2Vec (Mikolov et al., 2013) | 0.101 | 0.116 | 0.124 | 0.48 | 0.51 | 0.116 | 0.142 | 0.154 | 0.47 | 0.52 |
| GloVe (Pennington et al., 2014) | 0.146 | 0.175 | 0.169 | 0.50 | 0.49 | 0.131 | 0.170 | 0.168 | 0.50 | 0.49 |
| Fasttext (Bojanowski et al., 2017) | 0.180 | 0.200 | 0.191 | 0.51 | 0.48 | 0.146 | 0.196 | 0.191 | 0.50 | 0.49 |
| GN-GloVe (Zhao et al., 2018) | 0.032 | 0.055 | 0.054 | 0.50 | 0.49 | 0.024 | 0.085 | 0.088 | 0.49 | 0.50 |
| SBERT (Reimers and Gurevych, 2019) | 0.124 | 0.155 | 0.128 | 0.54 | 0.45 | 0.121 | 0.167 | 0.129 | 0.56 | 0.43 |
| SimCSE-RoBERTa (Gao et al., 2021) | 0.194 | 0.137 | 0.093 | 0.59 | 0.40 | 0.189 | 0.140 | 0.107 | 0.56 | 0.43 |
| InfoCSE-RoBERTa (Wu et al., 2022) | 0.199 | 0.222 | 0.211 | 0.51 | 0.48 | 0.228 | 0.265 | 0.241 | 0.52 | 0.47 |

Table 2: Result of Average Cosine Distance between gender and object in the COCO Captions and Flikr30K training datasets. The ratio is the gender bias rate **to**wards men/women. The results show there is a slight bias toward men. GloVe and GN-GloVe (balanced) show identical results on COCO Captions, which indicate that not all objects have a strong bias toward a specific gender. In particular, regarding non-biased objects, both models exhibit a low/similar bias ratio *e.g.* bicycle-gender (GloVe: m=0.31 | w=0.27, ratio=0.53) and (GN-GloVe: m=0.15 | w=0.13, ratio=0.53).

|  | Avg: Gender Score | | | Bias Ratio | | | Leakage | |
| Model | + person | + man | + woman | m | w | to-m | m | w |
|---|---|---|---|---|---|---|---|---|
| Human |  |  |  |  |  |  | 930 | 291 |
| **Gender Object Distance** | | | | | | | | |
| Transformer (Vaswani et al., 2017) | 0.075 | 0.062 | 0.033 | 0.82 | 0.44 | 0.65 | 0.85 | 1.40 |
| AoANet (Huang et al., 2019) | 0.079 | 0.061 | 0.047 | 0.77 | 0.59 | 0.56 | 0.82 | 1.29 |
| Vilbert (Lu et al., 2020) | 0.089 | 0.070 | 0.056 | 0.78 | 0.62 | 0.55 | 0.75 | 1.06 |
| OSCAR (Li et al., 2020) | 0.076 | 0.054 | 0.044 | 0.71 | 0.57 | 0.58 | 0.93 | 1.40 |
| BLIP (Li et al., 2022) | 0.068 | 0.050 | 0.040 | 0.73 | 0.58 | 0.55 | 0.83 | 1.32 |
| TraCLIPS-Reward (Cho et al., 2022) | 0.069 | 0.053 | 0.040 | 0.76 | 0.57 | 0.56 | 0.82 | 1.30 |
| BLIP-2 (Li et al., 2023) | 0.073 | 0.049 | 0.043 | 0.67 | 0.58 | 0.53 | 0.73 | 1.22 |
| **Gender Score** | | | | | | | | |
| Transformer (Vaswani et al., 2017) | 0.201 | 0.194 | 0.187 | 0.96 | 0.93 | 0.50 | 0.85 | 1.40 |
| AoANet (Huang et al., 2019) | 0.217 | 0.210 | 0.201 | 0.96 | 0.92 | 0.51 | 0.82 | 1.29 |
| Vilbert (Lu et al., 2020) | 0.225 | 0.217 | 0.207 | 0.96 | 0.92 | 0.51 | 0.75 | 1.06 |
| OSCAR (Li et al., 2020) | 0.196 | 0.188 | 0.181 | 0.95 | 0.92 | 0.50 | 0.93 | 1.40 |
| BLIP (Li et al., 2022) | 0.226 | 0.220 | 0.213 | 0.97 | 0.94 | 0.50 | 0.83 | 1.32 |
| TraCLIPS-Reward (Cho et al., 2022) | 0.198 | 0.194 | 0.185 | 0.97 | 0.93 | 0.51 | 0.82 | 1.30 |
| BLIP-2 (Li et al., 2023) | 0.224 | 0.215 | 0.210 | 0.95 | 0.93 | 0.50 | 0.73 | 1.22 |

Table 3: Result of the Average Cosine Distance and Gender Score, between gender and object information, on the Karpathy test split. Our score balances the bias better than direct Cosine Distance, primarily because not all objects exhibit strong gender bias. The leakage is a comparison between human-annotated caption and the classifier output that uses the gender-related object to influence the final gender prediction, such as associating women with food.

## 4 Gender Score

In this section, we describe the proposed **G**ender **S**core that estimates gender based on its semantic relation with the visual information extracted from the image. Sabir et al. (2022) proposed a caption re-ranking method that leverages visual semantic context. This approach utilized Belief Revision (Blok et al., 2003) to convert the similarity (*i.e.* Cosine Distance) into a probability measure.

**Belief Revision.** To obtain likelihood bias revisions based on similarity scores (*i.e.* gender, object), we need three parameters: (1) **Hypothesis (g)**: caption $y$ with the associated gender $a \in \{$man, woman$\}$, (2) **Informativeness (c)**: image object information $o$ confidence and (3) **Similarities**: the degree of relatedness between object and gender $sim(y, o)$.

$$\text{GS}_a(y) = \frac{1}{|\mathcal{D}|} \sum_{(y,o) \in \mathcal{D}} \text{P}\left(g_y \mid c_o\right) = \text{P}(g_y)^{\alpha} \quad (3)$$

where $\text{P}(g_y)$ is the *hypothesis* probability of $y$, $\mathcal{D}$ is the predicted captions with the gender $a$, and $\text{P}(c_o)$ is the probability of the evidence that causes hypothesis probability revision *i.e.* visual bias revision via object context $o$ from the image $I$, $o(I)$:

**Hypothesis**: $\text{P}(g_y)$ (caption $y$ with the gender $a$)
**Informativeness**: $1 - \text{P}(c_o)$ (object context $o$)
**Similarities**: $\alpha = \left[\frac{1 - \text{sim}(y,o)}{1 + \text{sim}(y,o)}\right]^{1 - \text{P}(c_o)}$ (visual bias)

The visual context $\text{P}(c_o)$ will revise the caption with the associated gender $\text{P}(g_y)$ (*i.e.* gender bias) if there is a semantic relation between them $sim(y, o)$. We discuss each component next:

**Hypothesis initial bias**: In visual-based belief revision, one of the conditions is to start with an initial hypothesis and then revise it using visual context and a similarity score. Therefore, we initialize the caption hypothesis $\text{P}(g_y)$ with a common observation $\text{P}(g_y')$, such as a Language Model (LM) (we

| Model | Bias Ratio Toward Men | | | | Bias Ratio Toward Women | | | |
| --- | --- | --- | --- | --- | --- | --- | --- | --- |
| | skateboard | kitchen | motorcycle | baseball | skateboard | kitchen | motorcycle | baseball |
| *Object Gender Co-Occ (Zhao et al., 2017)* | | | | | | | | |
| Transformer (Vaswani et al., 2017) | 0.96 | 0.50 | 0.83 | 0.75 | 0.05 | 0.50 | 0.16 | 0.25 |
| AoANet (Huang et al., 2019) | 0.97 | 0.51 | 0.85 | 0.81 | 0.02 | 0.48 | 0.14 | 0.18 |
| Vilbert (Lu et al., 2020) | 0.96 | 0.47 | 0.84 | 0.66 | 0.03 | 0.52 | 0.15 | 0.33 |
| OSCAR (Li et al., 2020) | 0.97 | 0.58 | 0.82 | 0.90 | 0.02 | 0.41 | 0.18 | 0.09 |
| BLIP (Li et al., 2022) | 0.96 | 0.52 | 0.88 | 0.97 | 0.03 | 0.47 | 0.11 | 0.02 |
| TraCLIPS-Reward (Cho et al., 2022) | 0.89 | 0.48 | 0.93 | 0.50 | 0.10 | 0.51 | 0.06 | 0.50 |
| BLIP-2 (Li et al., 2023) | 0.94 | 0.57 | 0.88 | 0.90 | 0.05 | 0.42 | 0.11 | 0.10 |
| *Gender Score* | | | | | | | | |
| Transformer (Vaswani et al., 2017) | 0.96 | 0.51 | 0.83 | 0.61 | 0.03 | 0.48 | 0.16 | 0.38 |
| AoANet (Huang et al., 2019) | 0.97 | 0.46 | 0.84 | 0.82 | 0.02 | 0.53 | 0.15 | 0.17 |
| Vilbert (Lu et al., 2020) | 0.96 | 0.53 | 0.84 | 0.65 | 0.03 | 0.46 | 0.15 | 0.34 |
| OSCAR (Li et al., 2020) | 0.98 | 0.42 | 0.78 | 0.83 | 0.01 | 0.57 | 0.21 | 0.16 |
| BLIP (Li et al., 2022) | 0.96 | 0.50 | 0.86 | 0.98 | 0.03 | 0.49 | 0.13 | 0.01 |
| TraCLIPS-Reward (Cho et al., 2022) | 0.88 | 0.43 | 0.92 | 0.50 | 0.11 | 0.56 | 0.07 | 0.49 |
| BLIP-2 (Li et al., 2023) | 0.93 | 0.56 | 0.82 | 0.89 | 0.06 | 0.43 | 0.17 | 0.10 |

Table 4: Example of the most common gender bias objects in COCO Captions, Karpathy test split. The result shows that our score (bias ratio) aligns closely with the existing Object Gender Co-Occ approach when applied to the most gender-biased objects toward men. Note that TraCLIPS-Reward (CLIPS+CIDEr) inherits biases from RL-CLIPS, resulting in distinct gender predictions and generates caption w/o a specific gender *i.e. person*, *baseball player*, etc.

consider this as an initial bias without visual). We employ (GPT-2) (Radford et al., 2019) with mean token probability since it achieves better results.

**Informativeness of bias information:** As the visual context probability $P(c_o)$ approaches 1 and in consequence is less informative (very frequent objects have no discriminative power since they may co-occur with any gender) $1 - P(c_o)$ approaches zero, causing $\alpha$ to get closer to 1, and thus, a smaller revision of $P(g'_y)$. Therefore, as we described in the Visual Context Information Section 2, we leverage a threshold and semantic filter visual context dataset from ResNet, Inception-ResNet v2 based Faster R-CNN object detector, and CLIP to extract top-$k$ textual visual context information from the image. The extracted object is used to measure the gender-to-object bias direct relation.

**Relatedness between hypothesis and bias information:** Likelihood revision occurs if there is a close correlation between the hypothesis and the new information. As the $sim(y, o)$ (gender, object), gets closer to 1 (higher relatedness) $\alpha$ gets closer to 0, and thus hypothesis probability is revised (*i.e.* gender bias) and raised closer to 1. Therefore, the initial hypothesis will be revised or backed off to 1 (no bias) based on the relatedness score. In our case, we employ Sentence-BERT to compute the Cosine Distance (Section 3) by using object $o$ as context for the caption $y$ with associated gender $a$.

## 5 Experiments

**Caption model.** We examine seven of the most recent Transformer state-of-the-art caption models: Transformer (Vaswani et al., 2017) with bottom-up top-down features (Anderson et al., 2018), AoANet (Huang et al., 2019), Vilbert (Lu et al., 2020), OSCAR (Li et al., 2020), BLIP (Li et al., 2022), Transformer with Reinforcement Learning (RL) CLIPS+CIDEr as image+text similarity Reward (TraCLIPS-Reward) (Cho et al., 2022) and Large $LM_{2.7B}$ based BLIP-2 (Li et al., 2023). Note that for a fair comparison with other pre-trained models, OSCAR uses a cross-entropy evaluation score rather than the RL-based CIDEr optimization score.

**Data.** Our gender bias score is performed on the ($82783 \times 5$ human annotations) COCO Captions dataset. For baselines (testing), the score is used to evaluate gender-to-object bias on the standard 5K Karpathy test split images (Karpathy and Fei-Fei, 2015) (GT is the average of five human bias ratios).

Our experiments apply visual semantics between the object or object with context *i.e.* caption and its related gender information to predict object-to-gender related bias. The proposed visual-gender scores are (1): Cosine Distance, which uses the similarity score to try to estimate the proper gender as shown in Table 2; (2) Gender Score, which carries out Belief Revision (visual bias likelihood revision) to revise the hypothesis initial bias via similarity *e.g.* Cosine Distance (gender, object). For our baseline, we adopt the Object Gender Co-Occ metric (Zhao et al., 2017) for the image captioning task.

In this work, we hypothesize that every image has a gender $\in$ {man, woman} or gender neutral (*i.e. person*), and our rationale is that each model suffers from *Right for the Wrong Reasons* (Hen-

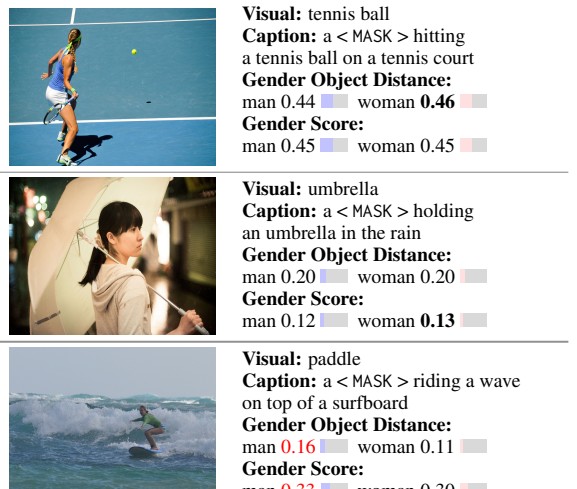

**Visual:** tennis ball
**Caption:** a < MASK > hitting a tennis ball on a tennis court
**Gender Object Distance:**
man 0.44    woman **0.46**
**Gender Score:**
man 0.45    woman 0.45

**Visual:** umbrella
**Caption:** a < MASK > holding an umbrella in the rain
**Gender Object Distance:**
man 0.20    woman 0.20
**Gender Score:**
man 0.12    woman **0.13**

**Visual:** paddle
**Caption:** a < MASK > riding a wave on top of a surfboard
**Gender Object Distance:**
man 0.16    woman 0.11
**Gender Score:**
man 0.33    woman 0.30

Figure 2: Examples of Gender Score Estimation and Cosine Distance. The result shows that (Top) the score balances the bias (as men and women have a similar bias for the sport *tennis*), (Middle) a slight bias object *umbrella* toward women, and (Bottom) men strong object bias relation (*paddle, surfboard*), the model adjusts the women bias while preserving the object gender score.

| Model | Gender | | Bias Ratio | |
| --- | --- | --- | --- | --- |
| | man | woman | to-m | to-w |
| Object Gender Co-Occ (Zhao et al., 2017) | | | | |
| Transformer | 792 | 408 | 0.66 | 0.34 |
| AoANet | 770 | 368 | 0.67 | 0.32 |
| Vilbert | 702 | 311 | 0.69 | 0.30 |
| OSCAR | 845 | 409 | 0.67 | 0.32 |
| BLIP | 775 | 385 | 0.66 | 0.33 |
| TraCLIPS-Reward | 769 | 381 | 0.66 | 0.33 |
| BLIP-2 | 695 | 356 | 0.66 | 0.33 |
| Gender Score (Gender Score Estimation) | | | | |
| Transformer | 616 | 217 | 0.73 | 0.26 |
| AoANet | 527 | 213 | 0.71 | 0.28 |
| Vilbert | 526 | 161 | 0.76 | 0.23 |
| OSCAR | 630 | 237 | 0.72 | 0.27 |
| BLIP | 554 | 240 | 0.69 | 0.30 |
| TraCLIPS-Reward | 537 | 251 | 0.68 | 0.31 |
| BLIP-2 | 498 | 239 | 0.67 | 0.32 |

Table 5: Comparison results between Object Gender Co-Occ (predicted gender-object results) and our gender score estimation on Karpathy test split. The proposed score measures gender bias more accurately, particularly when there is a strong object to gender bias relation.

dricks et al., 2018) or leakage (Wang et al., 2019) (see Table 3), such as associating all kitchens with women. Therefore, we want to explore all the cases and let the proposed distance/score decide which gender (*i.e.* bias) is in the image based on a visual bias. In particular, inspired by the cloze probability last word completion task (Gonzalez-Marquez, 2007), we generate two identical sentences but with a different gender, and then we compute the likelihood revisions between the sentence-gender and the caption using the object probability. Table 2 shows that GloVe and GN-GloVe (balanced) have identical results on COCO Captions dataset, which indicate that not all objects have a strong bias toward a specific gender. In addition, Table 3 shows that our score balances the bias of the Cosine Distance and demonstrates, on average, that not all objects have a strong gender bias. Also, our approach detects strong specific object-to-gender bias and has a similar result to the existing Object Gender Co-Occ method on the most biased object toward men, as shown in Table 4. TraCLIPS-Reward inherits biases from RL-CLIPS and thus generates caption w/o a specific gender (*e.g. person*, *guy*, etc). Therefore, we adopt the combined CLIPS+CIDEr Rewards, which suffer less gender prediction error.

**Gender Score Estimation.** In this experiment, we < MASK > the gender and use the object with context to predict the gender as shown in Figure 2. The idea

is to measure the amplified gender-to-object bias in pre-trained models. Table 5 shows that the fill-in gender score has a more bias towards men results than object-gender pair counting. The rationale is that the model can estimate the strong gender object bias as shown in Figure 2, including the false positive and leakage cases by the classifier.

**Discussion.** Our approach measures the amplified gender bias more accurately than the Object Gender Co-Occ (Zhao et al., 2017) in the following two scenarios: (1) where the gender is not obvious in the image and is misclassified by the caption baseline, and (2) when there is leakage by the classifier. In addition, unlike the Object Gender Co-Occ, our model balances the gender to object bias and only measures a strong object to gender bias relation as shown in Figure 2. For instance, the word "hitting" in a generated caption (as a stand-alone without context) is associated with a bias toward men more than women, and it will influence the final gender-to-caption bias score. However, our gender score balances the unwanted bias and only measures the pronounced gender to object bias relation.

## 6 Conclusion

In this work, we investigate the relation between objects and gender bias in image captioning systems. Our results show that not all objects exhibit a strong gender bias and only in special cases does an object have a strong gender bias. We also propose a Gender Score that can be used as an additional metric to the existing Object-Gender Co-Occ method.

## Limitations

The accuracy of our Gender Score heavily relies on the semantic relation (*i.e.* Cosine Distance) between gender and object, a high or low degree of similarity score between them can influence the final bias score negatively. Also, our model relies on a large pre-trained model(s), which inherently encapsulate their own latent biases that might impact the visual bias revision behavior in gender-to-object bias scenarios. Specifically, in cases when there are multiple strong bias contexts (*i.e.* non-objects context) with high similarity scores toward a particular gender present within the caption. This can imbalance the final gender-to-object score, leading to errors in gender prediction and bias estimation. In addition, the false positive object context information extracted by the visual classifier will result in inaccurate bias estimation.

## Ethical Considerations

We rely upon an existing range of well-known publicly available caption datasets crawled from the web and annotated by humans that assume a binary conceptualization of gender. Therefore, it is important to acknowledge that within the scope of this work, we are treating gender as strictly binary (*i.e.* man and woman) oversimplifies a complex and multifaceted aspect of human identity. Gender is better understood as a spectrum, with many variations beyond just two categories, and should be addressed in future work. Since all models are being trained on these datasets, we anticipate all models contain other biases (racial, cultural, *etc.*). For example, an observation in Table 1, when we remove gender bias, we notice the emergence of another bias, such as racial bias. For example, vectors representing *Black* person or women are closer together than those representing other colors, like *white*. Moreover, there is another form of bias that has received limited attention in the literature, the propagation of gender and racial bias via RL (*e.g.* RL-CLIPS). For instance, in Figure 1 the model associates gender with race, as "asian woman".

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
