# OpenReview forum: "Women Wearing Lipstick: Measuring the Bias Between an Object and Its Related Gender"
_EMNLP/2023/Conference — EMNLP 2023 Findings_

### Official Review · Reviewer_tUEX · 2023-07-22

**Soundness:** 3

**Excitement:**

3: Ambivalent: It has merits (e.g., it reports state-of-the-art results, the idea is nice), but there are key weaknesses (e.g., it describes incremental work), and it can significantly benefit from another round of revision. However, I won't object to accepting it if my co-reviewers champion it.

**Paper Topic And Main Contributions:**

The paper proposes a new gender bias metric called Gender Score for image captioning task.

**Questions For The Authors:**

Under what circumstances do you recommend one to use Gender Score instead of Object Gender Co-Occ? Or, when would one need Gender Score in addition to Object Gender Co-Occ?

**Reasons To Accept:**

The paper explores the use of cosine distance in measuring gender bias of image captioning and presents Gender Score as the new gender bias metric. The paper shows that the new method performs better than the prior art Object Gender Co-Occ as the measure of the gender bias of image captioning systems.

**Reasons To Reject:**

It wasn't clear to me the specific points that this new method complements the weaknesses of Object Gender Co-Occ. In Table 4, it is mentioned that the Gender Score measures the bias more accurately. It will be more clear to the reader how/why that might be possible. It was also unclear the tradeoffs one would make to choose Gender Score over Object Gender Co-Occ either.

UPDATE: Given that the authors will add a separate discussion section to go in more details of the comparison, this may be OK. Authors have provided a high level comparison in the response too.

**Reproducibility:**

4: Could mostly reproduce the results, but there may be some variation because of sample variance or minor variations in their interpretation of the protocol or method.

**Reviewer Confidence:**

3: Pretty sure, but there's a chance I missed something. Although I have a good feel for this area in general, I did not carefully check the paper's details, e.g., the math, experimental design, or novelty.

---

> ### Author Rebuttal · Authors · 2023-08-26
>
> We sincerely thank the reviewer for the reviews and question. Please find the response to your question below:
>
> - (Q1) Regarding the question: Under what circumstances do you recommend one to use Gender Score instead of Object Gender Co-Occ?
>
> (A1) Our approach measures the amplified gender bias more accurately than the Object Gender Co-Occ in the following two scenarios: (1) where the gender is not obvious in the image and is misclassified by the caption baseline, and (2) when there is leakage by the classifier. In addition, unlike the Object Gender Co-Occ, our model balances the gender to object bias (reduces the bias as shown in Figure 6 and Figure 7) and only measures a strong object to gender bias relation. Therefore, our approach can be used in the aforementioned scenarios as a stand-alone metric or as an additional metric alongside the existing Object Gender Co-Occ approach. The trade-off is that our model relies on a large pre-trained model(s), which carry their own inherent biases that might influence the bias score. However, as the reviewer recommended, we will add a dedicated discussion section for direct comparison and trade-off between the Gender Score and Object Gender Co-Occ.

---

### Official Review · Reviewer_EYYR · 2023-08-05

**Soundness:** 4

**Excitement:**

4: Strong: This paper deepens the understanding of some phenomenon or lowers the barriers to an existing research direction.

**Missing References:**

How does the method compare with other image segmentation metrics. For example, see ‘Gaussian correction for Adversarial learning of Boundaries.’ 2022 Signal Processing-Image Communication

**Paper Topic And Main Contributions:**

This paper looks at the bias between and object and its gender in a caption. They make use of pre-trained model Inception for object segmentation. They also use a pre-trained language model such as MASK.

**Reasons To Accept:**

Table 2 shows that the method balances bias better than the cosine distance metric.

Similarly, Table 4 also shows that the gender bias is more accurate especially for classifiers that misrecognise gender..

**Reasons To Reject:**

Pg 1, right col, line no 047, what is difference between gender of word and caption?

**Reproducibility:**

4: Could mostly reproduce the results, but there may be some variation because of sample variance or minor variations in their interpretation of the protocol or method.

**Reviewer Confidence:**

3: Pretty sure, but there's a chance I missed something. Although I have a good feel for this area in general, I did not carefully check the paper's details, e.g., the math, experimental design, or novelty.

---

> ### Author Rebuttal · Authors · 2023-08-26
>
> We sincerely thank the reviewer for the reviews and questions. Please find the response to your questions below:
>
> - (Q1,2) Regarding the questions: What is difference between gender of word and caption? and comparing our methods with other works?
>
> (A1,2)  In the gender-to-object approach, we investigate the amplified gender-to-object bias relation for image captioning at the word level and without the word context. In contrast, the gender-to-caption relation method delves into the bias relation at the sentence (caption) level with the surrounding word context that might influence the final gender bias score. For instance, the word “sitting” in a generated caption  (as a stand-alone without context) is associated with a bias toward “women” more than “men”, and it will influence the final gender-to-caption bias score. However, our gender score balances the unwanted bias and only measures the pronounced gender to object bias relation. Additionally, compared to other works, our semantic-based gender score can predict gender bias without unbalancing the dataset and can be easily incorporated into any off-the-shelf image captioning system.

---

### Official Review · Reviewer_vgnb · 2023-08-05

**Soundness:** 3

**Excitement:**

2: Mediocre: This paper makes marginal contributions (vs non-contemporaneous work), so I would rather not see it in the conference.

**Paper Topic And Main Contributions:**

This paper investigates the impact of objects on gender bias in image caption systems. The main problem addressed is the presence of gender bias in the generated captions, particularly when gender-specific objects are involved. The paper contributes to the understanding of gender bias in image captioning systems by investigating the gender object relation and proposing a visual semantic-based gender score. The gender score offers a practical tool for measuring bias and can be easily incorporated into image captioning models for improved evaluation and mitigation of gender bias.

**Reasons To Accept:**

1) The paper addresses an important and timely issue of gender bias in image caption systems, particularly focusing on the impact of gender-specific objects.
2) The introduction of a visual semantic-based gender score is a novel contribution that offers a practical and quantifiable metric for measuring the degree of gender bias in generated captions. This score can be easily integrated as a plug-in into existing image captioning systems, providing a standardized approach for bias evaluation.
3) The paper demonstrates the utility of the proposed gender score by showing its effectiveness in measuring the bias relationship between captions and gender. Moreover, the gender score serves as an additional metric to complement the existing Object Gender Co-Occ approach, enhancing the comprehensiveness of bias evaluation.

**Reasons To Reject:**

1) The main body of the paper appears too short, and several important content elements, such as the framework diagram and experimental control groups, should be included in the main text. Additionally, the paper lacks formal expressions for the mathematical equations, which could diminish the clarity and rigor of the presented methods.
2) Some tables have too small font sizes, and the lack of bottomlines reduces the readability and visual appeal of the paper. A clear presentation of results with appropriate font sizes and table formatting is essential for conveying findings effectively.
3) The paper only compares the proposed gender score with the cosine distance, without exploring more suitable or updated comparison methods， which would weaken the credibility and generalizability of the proposed gender score.

**Reproducibility:**

3: Could reproduce the results with some difficulty. The settings of parameters are underspecified or subjectively determined; the training/evaluation data are not widely available.

**Reviewer Confidence:**

3: Pretty sure, but there's a chance I missed something. Although I have a good feel for this area in general, I did not carefully check the paper's details, e.g., the math, experimental design, or novelty.

---

> ### Author Rebuttal · Authors · 2023-08-26
>
> We are grateful to the reviewer for the constructive comments. Please find the response to your comments below:
>
> - (Q1,2)  Regarding the comments about the framework diagram that should be included in the main text, small font sizes, and lack of formal expressions for the mathematical equations.
>
> (A1,2) We thank the reviewer for pointing this out. We will improve the presentation of the paper, and we will add the framework diagram to the main text from the appendix.
>
> - (Q3) Regarding the comment: the paper only compares the proposed gender score with the cosine distance.
>
> (A3) As the reviewer mentioned, our work only incorporates cosine distance in the proposed framework. However, the sole viable approach to compare against the likelihood revisions is the cosine distance (_i.e._ measuring the degree of the gender to object semantic relation bias before the object to gender bias likelihood revisions (gender score)). Additionally, in the attached appendix (Table 6 and Table 9), we report additional results comparing our approach against general baselines as a stand-alone, such as Language Model (GPT-2) and Mask Language Model (BERT).

---

### Meta-Review · Area_Chair_WDJY · 2023-09-19

**Recommendation:** 3

**Metareview:**

This paper proposes an analysis of how objects may be implicitly associated with genders in image captioning models. The analysis recovers traditional gendered associations, e.g., lipstick with women. The paper also presents a metric that can be used to evaluate gender bias in object representations. I would suggest adding some discussion of the limitations of studying gender as a binary category and of autoamtic gender classification in general.

---

### Decision · Program_Chairs · 2023-10-07

**Decision:**

Accept-Findings

**Comment:**

This paper proposes an analysis of how objects may be implicitly associated with genders in image captioning models. The analysis recovers traditional gendered associations, e.g., lipstick with women. The paper also presents a metric that can be used to evaluate gender bias in object representations. I would suggest adding some discussion of the limitations of studying gender as a binary category and of autoamtic gender classification in general.